# LEARNING TO COUNT OBJECTS IN NATURAL IMAGES FOR VISUAL QUESTION ANSWERING

**Yan Zhang & Jonathon Hare & Adam Prügel-Bennett**
Department of Electronics and Computer Science
University of Southampton
{yz5n12,jsh2,apb}@ecs.soton.ac.uk

## ABSTRACT

Visual Question Answering (VQA) models have struggled with counting objects in natural images so far. We identify a fundamental problem due to soft attention in these models as a cause. To circumvent this problem, we propose a neural network component that allows robust counting from object proposals. Experiments on a toy task show the effectiveness of this component and we obtain state-of-the-art accuracy on the number category of the VQA v2 dataset without negatively affecting other categories, even outperforming ensemble models with our single model. On a difficult balanced pair metric, the component gives a substantial improvement in counting over a strong baseline by 6.6%.

## 1 INTRODUCTION

Consider the problem of counting how many cats there are in Figure 1. Solving this involves several rough steps: understanding what instances of that type can look like, finding them in the image, and adding them up. This is a common task in Visual Question Answering (VQA) – answering questions about images – and is rated as among the tasks requiring the lowest human age to be able to answer (Antol et al., 2015). However, current models for VQA on natural images struggle to answer any counting questions successfully outside of dataset biases (Jabri et al., 2016).

One reason for this is the presence of a fundamental problem with counting in the widely-used soft attention mechanisms (section 3). Another reason is that unlike standard counting tasks, there is no ground truth labeling of where the objects to count are. Coupled with the fact that models need to be able to count a large variety of objects and that, ideally, performance on non-counting questions should not be compromised, the task of counting in VQA seems very challenging.

To make this task easier, we can use object proposals – pairs of a bounding box and object features – from object detection networks as input instead of learning from pixels directly. In any moderately complex scene, this runs into the issue of double-counting overlapping object proposals. This is a problem present in many natural images, which leads to inaccurate counting in real-world scenarios.

Our main contribution is a differentiable neural network component that tackles this problem and consequently can learn to count (section 4). Used alongside an attention mechanism, this component avoids a fundamental limitation of soft attention while producing strong counting features. We provide experimental evidence of the effectiveness of this component (section 5). On a toy dataset, we demonstrate that this component enables robust counting in a variety of scenarios. On the number category of the VQA v2 Open-Ended dataset (Goyal et al., 2017), a relatively simple baseline model using the counting component outperforms all previous models – including large ensembles of state-of-the-art methods – without degrading performance on other categories. [1]

## 2 RELATED WORK

Usually, greedy non-maximum suppression (NMS) is used to eliminate duplicate bounding boxes. The main problem with using it as part of a model is that its gradient is piecewise constant. Various

---

[1] Our implementation is available at https://github.com/Cyanogenoid/vqa-counting.

differentiable variants such as by Azadi et al. (2017), Hosang et al. (2017), and Henderson & Ferrari (2017) exist. The main difference is that, since we are interested in counting, our component does not need to make discrete decisions about which bounding boxes to keep; it outputs counting features, not a smaller set of bounding boxes. Our component is also easily integrated into standard VQA models that utilize soft attention without any need for other network architecture changes and can be used without using true bounding boxes for supervision.

On the VQA v2 dataset (Goyal et al., 2017) that we apply our method on, only few advances on counting questions have been made. The main improvement in accuracy is due to the use of object proposals in the visual processing pipeline, proposed by Anderson et al. (2017). Their object proposal network is trained with classes in singular and plural forms, for example "tree" versus "trees", which only allows primitive counting information to be present in the object features after region-of-interest pooling. Our approach differs in the way that instead of relying on counting features being present in the input, we create counting features using information present in the attention map over object proposals. This has the benefit of being able to count anything that the attention mechanism can discriminate instead of only objects that belong to the predetermined set of classes that had plural forms.

Using these object proposals, Trott et al. (2018) train a sequential counting mechanism with a reinforcement learning loss on the counting question subsets of VQA v2 and Visual Genome. They achieve a small increase in accuracy and can obtain an interpretable set of objects that their model counted, but it is unclear whether their method can be integrated into traditional VQA models due to their loss not applying to non-counting questions. Since they evaluate on their own dataset, their results can not be easily compared to existing results in VQA.

Methods such as by Santoro et al. (2017) and Perez et al. (2017) can count on the synthetic CLEVR VQA dataset (Johnson et al., 2017) successfully without bounding boxes and supervision of where the objects to count are. They also use more training data (∼250,000 counting questions in the CLEVR training set versus ∼50,000 counting questions in the VQA v2 training set), much simpler objects, and synthetic question structures.

More traditional approaches based on Lempitsky & Zisserman (2010) learn to produce a target density map, from which a count is computed by integrating over it. In this setting, Cohen et al. (2017) make use of overlaps of convolutional receptive fields to improve counting performance. Chattopadhyay et al. (2017) use an approach that divides the image into smaller non-overlapping chunks, each of which is counted individually and combined together at the end. In both of these contexts, the convolutional receptive fields or chunks can be seen as sets of bounding boxes with a fixed structure in their positioning. Note that while Chattopadhyay et al. (2017) evaluate their models on a small subset of counting questions in VQA, major differences in training setup make their results not comparable to our work.

## 3 PROBLEMS WITH SOFT ATTENTION

The main message in this section is that using the feature vectors obtained after the attention mechanism is not enough to be able to count; the attention maps themselves should be used, which is what we do in our counting component.

Models in VQA have consistently benefited from the use of soft attention (Mnih et al., 2014; Bahdanau et al., 2015) on the image, commonly implemented with a shallow convolutional network. It learns to output a weight for the feature vector at each spatial position in the feature map, which is first normalized and then used for performing a weighted sum over the spatial positions to produce a single feature vector. However, soft spatial attention severely limits the ability for a model to count.

Consider the task of counting the number of cats for two images: an image showing a single cat on a clean background and an image that consists of two side-by-side copies of the first image. What we will describe applies to both spatial feature maps and sets of object proposals as input, but we focus on the latter case for simplicity. With an object detection network, we detect one cat in the first image and two cats in the second image, producing the same feature vector for all three detections. The attention mechanism then assigns all three instances of the same cat the same weight.

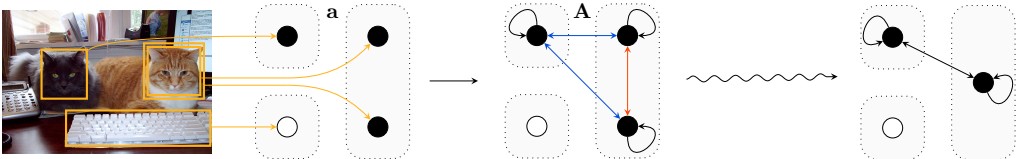

Figure 1: Simplified example about counting the number of cats. The light-colored cat is detected twice and results in a duplicate proposal. This shows the conversion from the attention weights **a** to a graph representation **A** and the eventual goal of this component with exactly one proposal per true object. There are 4 proposals (vertices) capturing 3 underlying objects (groups in dotted lines). There are 3 relevant proposals (black with weight 1) and 1 irrelevant proposal (white with weight 0). Red edges mark *intra*-object edges between duplicate proposals and blue edges mark the main *inter*-object duplicate edges. In graph form, the object groups, coloring of edges, and shading of vertices serve illustration purposes only; the model does not have these access to these directly.

The usual normalization used for the attention weights is the softmax function, which normalizes the weights to sum to 1. Herein lies the problem: the cat in the first image receives a normalized weight of 1, but the two cats in the second image now each receive a weight of 0.5. After the weighted sum, we are effectively averaging the two cats in the second image back to a single cat. As a consequence, the feature vector obtained after the weighted sum is exactly the same between the two images and we have lost all information about a possible count from the attention map. Any method that normalizes the weights to sum to 1 suffers from this issue.

Multiple glimpses (Larochelle & Hinton, 2010) – sets of attention weights that the attention mechanism outputs – or several steps of attention (Yang et al., 2016; Lu et al., 2016) do not circumvent this problem. Each glimpse or step can not separate out an object each, since the attention weight given to one feature vector does not depend on the other feature vectors to be attended over. Hard attention (Ba et al., 2015; Mnih et al., 2014) and structured attention (Kim et al., 2017) may be possible solutions to this, though no significant improvement in counting ability has been found for the latter so far (Zhu et al., 2017). Ren & Zemel (2017) circumvent the problem by limiting attention to only work within one bounding box at a time, remotely similar to our approach of using object proposal features.

Without normalization of weights to sum to one, the scale of the output features depends on the number of objects detected. In an image with 10 cats, the output feature vector is scaled up by 10. Since deep neural networks are typically very scale-sensitive – the scale of weight initializations and activations is generally considered quite important (Mishkin & Matas, 2016) – and the classifier would have to learn that joint scaling of all features is somehow related to count, this approach is not reasonable for counting objects. This is evidenced in Teney et al. (2017) where they provide evidence that sigmoid normalization not only degrades accuracy on non-number questions slightly, but also does not help with counting.

## 4 COUNTING COMPONENT

In this section, we describe a differentiable mechanism for counting from attention weights, while also dealing with the problem of overlapping object proposals to reduce double-counting of objects. This involves some nontrivial details to produce counts that are as accurate as possible. The main idea is illustrated in Figure 1 with the two main steps shown in Figure 2 and Figure 3. The use of this component allows a model to count while still being able to exploit the benefits of soft attention.

Our key idea for dealing with overlapping object proposals is to turn these object proposals into a graph that is based on how they overlap. We then remove and scale edges in a specific way such that an estimate of the number of underlying objects is recovered.

Our general strategy is to primarily design the component for the unrealistic extreme cases of perfect attention maps and bounding boxes that are either fully overlapping or fully distinct. By introducing some parameters and only using differentiable operations, we give the ability for the module to interpolate between the correct behaviours for these extreme cases to handle the more realistic cases.

These parameters are responsible for handling variations in attention weights and partial bounding box overlaps in a manner suitable for a given dataset.

To achieve this, we use several piecewise linear functions $f_1, \ldots, f_8$ as activation functions (defined in Appendix A), approximating arbitrary functions with domain and range [0, 1]. The shapes of these functions are learned to handle the specific nonlinear interactions necessary for dealing with overlapping proposals. Through their parametrization we enforce that $f_k(0) = 0$, $f_k(1) = 1$, and that they are monotonically increasing. The first two properties are required so that the extreme cases that we explicitly handle are left unchanged. In those cases, $f_k$ is only applied to values of 0 or 1, so the activation functions can be safely ignored for understanding how the component handles them. By enforcing monotonicity, we can make sure that, for example, an increased value in an attention map should never result in the prediction of the count to decrease.

## 4.1 INPUT

Given a set of features from object proposals, an attention mechanism produces a weight for each proposal based on the question. The counting component takes as input the $n$ largest attention weights $\mathbf{a} = [a_1, \ldots, a_n]^\mathsf{T}$ and their corresponding bounding boxes $\mathbf{b} = [b_1, \ldots, b_n]^\mathsf{T}$. We assume that the weights lie in the interval [0, 1], which can easily be achieved by applying a logistic function.

In the extreme cases that we explicitly handle, we assume that the attention mechanism assigns a value of 1 to $a_i$ whenever the $i$th proposal contains a relevant object and a value of 0 whenever it does not. This is in line with what usual soft attention mechanisms learn, as they produce higher weights for relevant inputs. We also assume that either two object proposals fully overlap (in which case they must be showing the same object and thus receive the same attention weight) or that they are fully distinct (in which case they show different objects). Keep in mind that while we make these assumptions to make reasoning about the behaviour easier, the learned parameters in the activation functions are intended to handle the more realistic scenarios when the assumptions do not apply.

Instead of partially overlapping proposals, the problem now becomes the handling of *exact duplicate* proposals of underlying objects in a differentiable manner.

## 4.2 DEDUPLICATION

We start by changing the vector of attention weights $\mathbf{a}$ into a graph representation in which bounding boxes can be utilized more easily. Hence, we compute the outer product of the attention weights to obtain an attention matrix.

$$\mathbf{A} = \mathbf{a}\mathbf{a}^\mathsf{T} \qquad (1)$$

$\mathbf{A} \in \mathbb{R}^{n \times n}$ can be interpreted as an adjacency matrix for a weighted directed graph. In this graph, the $i$th vertex represents the object proposal associated with $a_i$ and the edge between any pair of vertices $(i, j)$ has weight $a_i a_j$. In the extreme case where $a_i$ is virtually 0 or 1, products are equivalent to logical AND operators. It follows that the subgraph containing only the vertices satisfying $a_i = 1$ is a complete digraph with self-loops.

In this representation, our objective is to eliminate edges in such a way that, conceptually, the underlying true objects – instead of proposals thereof – are the vertices of that complete subgraph. In order to then turn that graph into a count, recall that the number of edges $|E|$ in a complete digraph with self-loops relates to the number of vertices $|V|$ through $|E| = |V|^2$. $|E|$ can be computed by summing over the entries in an adjacency matrix and $|V|$ is then the count. Notice how when $|E|$ is set to the sum over $\mathbf{A}$, $\sqrt{|E|} = \sum_i a_i$ holds. This convenient property implies that when all proposals are fully distinct, the component can output the same as simply summing over the original attention weights by default.

There are two types of duplicate edges to eliminate to achieve our objective: *intra*-object edges and *inter*-object edges.

### 4.2.1 INTRA-OBJECT EDGES

First, we eliminate intra-object edges between duplicate proposals of a single underlying object.

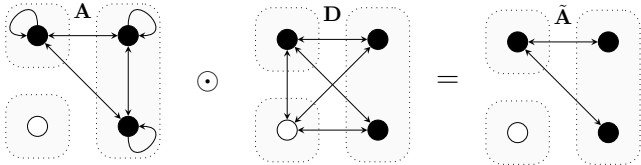

Figure 2: Removal of intra-object edges by masking the edges of the attention matrix $\mathbf{A}$ with the distance matrix $\mathbf{D}$. The black vertices now form a graph without self-loops. The self-loops need to be added back in later.

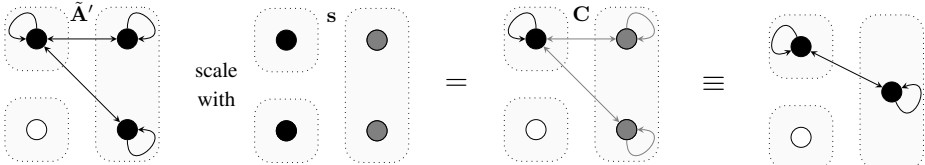

Figure 3: Removal of duplicate inter-object edges by computing a scaling factor for each vertex and scaling $\tilde{\mathbf{A}}'$ accordingly. $\tilde{\mathbf{A}}'$ is $\tilde{\mathbf{A}}$ with self-loops already added back in. The scaling factor for one vertex is computed by counting how many vertices have outgoing edges to the same set of vertices; all edges of the two proposals on the right are scaled by $0.5$. This can be seen as averaging proposals within each object and is equivalent to removing duplicate proposals altogether under a sum.

To compare two bounding boxes, we use the usual intersection-over-union (IoU) metric. We define the distance matrix $\mathbf{D} \in \mathbb{R}^{n \times n}$ to be

$$D_{ij} = 1 - \text{IoU}(b_i, b_j) \tag{2}$$

$\mathbf{D}$ can also be interpreted as an adjacency matrix. It represents a graph that has edges everywhere except when the two bounding boxes that an edge connects would overlap.

Intra-object edges are removed by elementwise multiplying ($\odot$) the distance matrix with the attention matrix (Figure 2).

$$\tilde{\mathbf{A}} = f_1(\mathbf{A}) \odot f_2(\mathbf{D}) \tag{3}$$

$\tilde{\mathbf{A}}$ no longer has self-loops, so we need to add them back in at a later point to still satisfy $|E| = |V|^2$. Notice that we start making use of the activation functions mentioned earlier to handle intermediate values in the interval $(0, 1)$ for both $\mathbf{A}$ and $\mathbf{D}$. They regulate the influence of attention weights that are not close to 0 or 1 and the influence of partial overlaps.

### 4.2.2 INTER-OBJECT EDGES

Second, we eliminate inter-object edges between duplicate proposals of different underlying objects.

The main idea (depicted in Figure 3) is to count the number of proposals associated to each invidual object, then scale down the weight of their associated edges by that number. If there are two proposals of a single object, the edges involving those proposals should be scaled by 0.5. In essence, this averages over the proposals within each underlying object because we only use the sum over the edge weights to compute the count at the end. Conceptually, this reduces multiple proposals of an object down to one as desired. Since we do not know how many proposals belong to an object, we have to estimate this. We do this by using the fact that proposals of the same object are similar.

Keep in mind that $\tilde{\mathbf{A}}$ has no self-loops nor edges between proposals of the same object. As a consequence, two nonzero rows in $\tilde{\mathbf{A}}$ are the same if and only if the proposals are the same. If the two rows differ in at least one entry, then one proposal overlaps a proposal that the other proposal does not overlap, so they must be different proposals. This means for comparing rows, we need a

similarity function that satisfies the criteria of taking the value $1$ when they differ in no places and $0$ if they differ in at least one place. We define a differentiable similarity between proposals $i$ and $j$ as

$$\text{Sim}_{ij} = f_3(1 - |a_i - a_j|) \prod_k f_3(1 - |X_{ik} - X_{jk}|) \tag{4}$$

where $\mathbf{X} = f_4(\mathbf{A}) \odot f_5(\mathbf{D})$ is the same as $\tilde{\mathbf{A}}$ except with different activation functions. The $\prod$ term compares the rows of proposals $i$ and $j$. Using this term instead of $f_4(1 - D_{ij})$ was more robust to inaccurate bounding boxes in initial experiments.

Note that the $f_3(1 - |a_i - a_j|)$ term handles the edge case when there is only one proposal to count. Since $\mathbf{X}$ does not have self-loops, $\mathbf{X}$ contains only zeros in that case, which causes the row corresponding to $a_i = 1$ to be incorrectly similar to the rows where $a_{j \neq i} = 0$. By comparing the attention weights through that term as well, this issue is avoided.

Now that we can check how similar two proposals are, we count the number of times any row is the same as any other row and compute a scaling factor $s_i$ for each vertex $i$.

$$s_i = 1 / \sum_j \text{Sim}_{ij} \tag{5}$$

The time complexity of computing $\mathbf{s} = [s_1, \ldots, s_n]^\mathsf{T}$ is $\Theta(n^3)$ as there are $n^2$ pairs of rows and $\Theta(n)$ operations to compute the similarity of any pair of rows.

Since these scaling factors apply to each vertex, we have to expand $\mathbf{s}$ into a matrix using the outer product in order to scale both incoming and outgoing edges of each vertex. We can also add self-loops back in, which need to be scaled by $\mathbf{s}$ as well. Then, the count matrix $\mathbf{C}$ is

$$\mathbf{C} = \tilde{\mathbf{A}} \odot \mathbf{s}\mathbf{s}^\mathsf{T} + \text{diag}(\mathbf{s} \odot f_1(\mathbf{a} \odot \mathbf{a})) \tag{6}$$

where $\text{diag}(\cdot)$ expands a vector into a diagonal matrix with the vector on the diagonal.

The scaling of self-loops involves a non-obvious detail. Recall that the diagonal that was removed when going from $\mathbf{A}$ to $\tilde{\mathbf{A}}$ contains the entries $f_1(\mathbf{a} \odot \mathbf{a})$. Notice however that we are scaling this diagonal by $\mathbf{s}$ and not $\mathbf{s} \odot \mathbf{s}$. This is because the number of inter-object edges scales quadratically with respect to the number of proposals per object, but the number of self-loops only scales linearly.

## 4.3 Output

Under a sum, $\mathbf{C}$ is now equivalent to a complete graph with self-loops that involves all relevant objects instead of relevant proposals as originally desired.

To turn $\mathbf{C}$ into a count $c$, we set $|E| = \sum_{i,j} C_{ij}$ as mentioned and

$$c = |V| = \sqrt{|E|} \tag{7}$$

We verified experimentally that when our extreme case assumptions hold, $c$ is always an integer and equal to the correct count, regardless of the number of duplicate object proposals.

To avoid issues with scale when the number of objects is large, we turn this single feature into several classes, one for each possible number. Since we only used the object proposals with the largest $n$ weights, the predicted count $c$ can be at most $n$. We define the output $\mathbf{o} = [o_0, o_1, \ldots, o_n]^\mathsf{T}$ to be

$$o_i = \max(0, 1 - |c - i|) \tag{8}$$

This results in a vector that is $1$ at the index of the count and $0$ everywhere else when $c$ is exactly an integer, and a linear interpolation between the two corresponding one-hot vectors when the count falls inbetween two integers.

### 4.3.1 OUTPUT CONFIDENCE

Finally, we might consider a prediction made from values of $\mathbf{a}$ and $\mathbf{D}$ that are either close to $0$ or close to $1$ to be more reliable – we explicitly handle these after all – than when many values are close to $0.5$. To incorporate this idea, we scale $\mathbf{o}$ by a confidence value in the interval $[0, 1]$.

We define $p_\mathbf{a}$ and $p_\mathbf{D}$ to be the average distances to $0.5$. The choice of $0.5$ is not important, because the module can learn to change it by changing where $f_6(x) = 0.5$ and $f_7(x) = 0.5$.

$$p_\mathbf{a} = \frac{1}{n} \sum_i |f_6(a_i) - 0.5| \tag{9}$$

$$p_\mathbf{D} = \frac{1}{n^2} \sum_{i,j} |f_7(D_{ij}) - 0.5| \tag{10}$$

Then, the output of the component with confidence scaling is

$$\tilde{\mathbf{o}} = f_8(p_\mathbf{a} + p_\mathbf{D}) \cdot \mathbf{o} \tag{11}$$

In summary, we only used diffentiable operations to deduplicate object proposals and obtain a feature vector that represents the predicted count. This allows easy integration into any model with soft attention, enabling a model to count from an attention map.

## 5 EXPERIMENTS

### 5.1 TOY TASK

First, we design a simple toy task to evaluate counting ability. This dataset is intended to only evaluate the performance of counting; thus, we skip any processing steps that are not directly related such as the processing of an input image. Samples from this dataset are given in Appendix D

The classification task is to predict an integer count $\hat{c}$ of true objects, uniformly drawn from 0 to 10 inclusive, from a set of bounding boxes and the associated attention weights. 10 square bounding boxes with side length $l \in (0, 1]$ are placed in a square image with unit side length. The x and y coordinates of their top left corners are uniformly drawn from $U(0, 1 - l)$ so that the boxes do not extend beyond the image border. $l$ is used to control the overlapping of bounding boxes: a larger $l$ leads to the fixed number of objects to be more tightly packed, increasing the chance of overlaps. $\hat{c}$ number of these boxes are randomly chosen to be true bounding boxes. The *score* of a bounding box is the maximum IoU overlap of it with any true bounding box. Then, the attention weight is a linear interpolation between the score and a noise value drawn from $U(0, 1)$, with $q \in [0, 1]$ controlling this trade-off. $q$ is the attention noise parameter: when $q$ is 0, there is no noise and when $q$ is 1, there is no signal. Increasing $q$ also indirectly simulates imprecise placements of bounding boxes.

We compare the counting component against a simple baseline that simply sums the attention weights and turns the sum into a feature vector with Equation 8. Both models are followed by a linear projection to the classes 0 to 10 inclusive and a softmax activation. They are trained with cross-entropy loss for 1000 iterations using Adam (Kingma & Ba, 2015) with a learning rate of 0.01 and a batch size of 1024.

### 5.1.1 RESULTS

The results of varying $l$ while keeping $q$ fixed at various values and vice versa are shown in Figure 4. Regardless of $l$ and $q$, the counting component performs better than the baseline in most cases, often significantly so. Particularly when the noise is low, the component can deal with high values for $l$ very successfully, showing that it accomplishes the goal of increased robustness to overlapping proposals. The component also handles moderate noise levels decently as long as the overlaps are limited. The performance when both $l$ and $q$ are high is closely matched by the baseline, likely due to the high difficulty of those parametrizations leaving little information to extract in the first place.

We can also look at the shape of the activation functions themselves, shown in Figure 5 and Appendix C, to understand how the behaviour changes with varying dataset parameters. For simplicity,

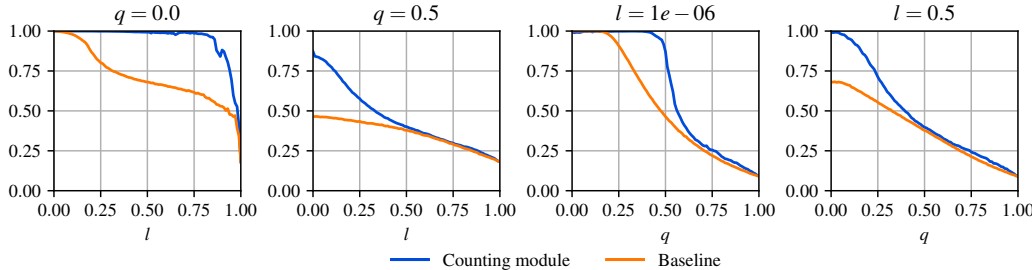

Figure 4: Accuracies on the toy task as side length $l$ and noise $q$ are varied in 0.01 step sizes.

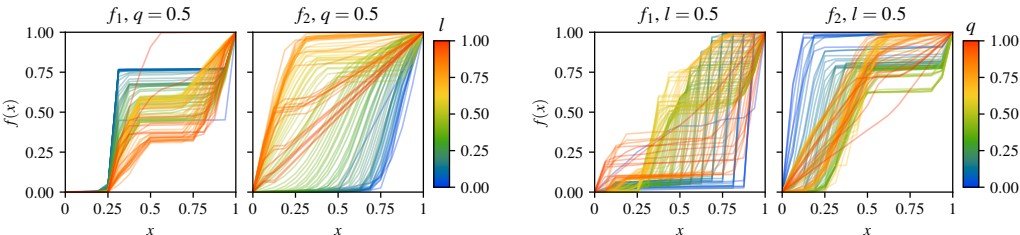

Figure 5: Shapes of trained activation functions $f_1$ (attention weights) and $f_2$ (bounding box distances) for varying bounding box side lengths (left) or the noise (right) in the dataset, varied in 0.01 step sizes. Best viewed in color.

we limit our description to the two easiest-to-interpret functions: $f_1$ for the attention weights and $f_2$ for the bounding box distances.

When increasing the side length, the height of the "step" in $f_1$ decreases to compensate for the generally greater degree of overlapping bounding boxes. A similar effect is seen with $f_2$: it varies over requiring a high pairwise distance when $l$ is low – when partial overlaps are most likely spurious – and considering small distances enough for proposals to be considered different when $l$ is high. At the highest values for $l$, there is little signal in the overlaps left since everything overlaps with everything, which explains why $f_2$ returns to its default linear initialization for those parameters.

When varying the amount of noise, without noise $f_1$ resembles a step function where the step starts close to $x = 1$ and takes a value of close to 1 after the step. Since a true proposal will always have a weight of 1 when there is no noise, anything below this can be safely zeroed out. With increasing noise, this step moves away from 1 for both $x$ and $f_1(x)$, capturing the uncertainty when a bounding box belongs to a true object. With lower $q$, $f_2$ considers a pair of proposals to be distinct for lower distances, whereas with higher $q$, $f_2$ follows a more sigmoidal shape. This can be explained by the model taking the increased uncertainty of the precise bounding box placements into account by requiring higher distances for proposals to be considered completely different.

## 5.2 VQA

VQA v2 (Goyal et al., 2017) is the updated version of the VQA v1 dataset (Antol et al., 2015) where greater care has been taken to reduce dataset biases through balanced pairs: for each question, a pair of images is identified where the answer to that question differs. The standard accuracy metric on this dataset accounts for disagreements in human answers by averaging $\min(\frac{1}{3} \text{ agreeing}, 1)$ over all 10-choose-9 subsets of human answers, where *agreeing* is the number of human answers that agree with the given answer. This can be shown to be equal to $\min(0.3 \text{ agreeing}, 1)$ without averaging.

We use an improved version of the strong VQA baseline by Kazemi & Elqursh (2017) as baseline model (details in Appendix B). We have not performed any tuning of this baseline to maximize the performance difference between it and the baseline with counting module. To augment this model with the counting component, we extract the attention weights of the first attention glimpse (there are two in the baseline) before softmax normalization, and feed them into the counting component after

Table 1: Results on VQA v2 of the top models along with our results. Entries marked with (Ens.) are ensembles of models. At the time of writing, our model with the counting module places third among all entries. All models listed here use object proposal features and are trained on the training and validation sets. The top-performing ensemble models use additional pre-trained word embeddings, which we do not use.

| | VQA v2 test-dev | | | | VQA v2 test | | | |
|---|---|---|---|---|---|---|---|---|
| Model | Yes/No | **Number** | Other | All | Yes/No | **Number** | Other | All |
| Teney et al. (2017) | 81.82 | 44.21 | 56.05 | 65.32 | 82.20 | 43.90 | 56.26 | 65.67 |
| Teney et al. (2017) (Ens.) | 86.08 | 48.99 | 60.80 | 69.87 | 86.60 | 48.64 | 61.15 | 70.34 |
| Zhou et al. (2017) | 84.27 | 49.56 | 59.89 | 68.76 | – | – | – | – |
| Zhou et al. (2017) (Ens.) | – | – | – | – | 86.65 | 51.13 | 61.75 | 70.92 |
| Baseline | 82.98 | 46.88 | 58.99 | 67.50 | 83.21 | 46.60 | 59.20 | 67.78 |
| + counting module | 83.14 | **51.62** | 58.97 | 68.09 | 83.56 | **51.39** | 59.11 | 68.41 |

Table 2: Results on the VQA v2 validation set with models trained only on the training set. Reported are the mean accuracies and sample standard deviations ($\pm$) over 4 random initializations.

| | VQA accuracy | | | Balanced pair accuracy | | |
|---|---|---|---|---|---|---|
| Model | Number | **Count** | All | Number | **Count** | All |
| Baseline | 44.83$\pm$0.2 | 51.69$\pm$0.2 | 64.80$\pm$0.0 | 17.34$\pm$0.2 | 20.02$\pm$0.2 | 36.44$\pm$0.1 |
| + NMS | 44.60$\pm$0.1 | 51.41$\pm$0.1 | 64.80$\pm$0.1 | 17.06$\pm$0.1 | 19.72$\pm$0.1 | 36.44$\pm$0.2 |
| + counting module | 49.36$\pm$0.1 | **57.03**$\pm$0.0 | 65.42$\pm$0.1 | 23.10$\pm$0.2 | **26.63**$\pm$0.2 | 37.19$\pm$0.1 |

applying a logistic function. Since object proposal features from Anderson et al. (2017) vary from 10 to 100 per image, a natural choice for the number of top-$n$ proposals to use is 10. The output of the component is linearly projected into the same space as the hidden layer of the classifier, followed by ReLU activation, batch normalization, and addition with the features in the hidden layer.

### 5.2.1 RESULTS

Table 1 shows the results on the official VQA v2 leaderboard. The baseline with our component has a significantly higher accuracy on number questions without compromising accuracy on other categories compared to the baseline result. Despite our single-model baseline being substantially worse than the state-of-the-art, by simply adding the counting component we outperform even the 8-model ensemble in Zhou et al. (2017) on the number category. We expect further improvements in number accuracy when incorporating their techniques to improve the quality of attention weights, especially since the current state-of-the-art models suffer from the problems with counting that we mention in section 3. Some qualitative examples of inputs and activations within the counting component are shown in Appendix E.

We also evaluate our models on the validation set of VQA v2, shown in Table 2. This allows us to consider only the counting questions within number questions, since number questions include questions such as "what time is it?" as well. We treat any question starting with the words "how many" as a counting question. As we expect, the benefit of using the counting module on the counting question subset is higher than on number questions in general. Additionally, we try an approach where we simply replace the counting module with NMS, using the average of the attention glimpses as scoring, and one-hot encoding the number of proposals left. The NMS-based approach, using an IoU threshold of 0.5 and no score thresholding based on validation set performance, does not improve on the baseline, which suggests that the piecewise gradient of NMS is a major problem for learning to count in VQA and that conversely, there is a substantial benefit to being able to differentiate through the counting module.

Additionally, we can evaluate the accuracy over balanced pairs as proposed by Teney et al. (2017): the ratio of balanced pairs on which the VQA accuracy for both questions is 1.0. This is a much

more difficult metric, since it requires the model to find the subtle details between images instead of being able to rely on question biases in the dataset. First, notice how all balanced pair accuracies are greatly reduced compared to their respective VQA accuracy. More importantly, the absolute accuracy improvement of the counting module is still fully present with the more challenging metric, which is further evidence that the component can properly count rather than simply fitting better to dataset biases.

When looking at the activation functions of the trained model, shown in Figure 9, we find that some characteristics of them are shared with high-noise parametrizations of the toy dataset. This suggests that the current attention mechanisms and object proposal network are still very inaccurate, which explains the perhaps small-seeming increase in counting performance. This provides further evidence that the balanced pair accuracy is maybe a more reflective measure of how well current VQA models perform than the overall VQA accuracies of over 70% of the current top models.

## 6 CONCLUSION

After understanding why VQA models struggle to count, we designed a counting component that alleviates this problem through differentiable bounding box deduplication. The component can readily be used alongside any future improvements in VQA models, as long as they still use soft attention as all current top models on VQA v2 do. It has uses outside of VQA as well: for many counting tasks, it can allow an object-proposal-based approach to work without ground-truth objects available as long as there is a – possibly learned – per-proposal scoring (for example using a classification score) and a notion of how dissimilar a pair of proposals are. Since each step in the component has a clear purpose and interpretation, the learned weights of the activation functions are also interpretable. The design of the counting component is an example showing how by encoding inductive biases into a deep learning model, challenging problems such as counting of arbitrary objects can be approached when only relatively little supervisory information is available.

For future research, it should be kept in mind that VQA v2 requires a versatile skill set that current models do not have. To make progress on this dataset, we advocate focusing on *understanding* of what the current shortcomings of models are and finding ways to mitigate them.

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

## A    PIECEWISE LINEAR ACTIVATION FUNCTION

Intuitively, the interval $[0, 1]$ is split into $d$ equal size intervals. Each contains a line segment that is connected to the neighboring line segments at the boundaries of the intervals. These line segments form the shape of the activation function.

For each function $f_k$, there are $d$ weights $w_{k1}, \ldots, w_{kd}$, where the weight $w_{ki}$ is the gradient for the interval $[\frac{i-1}{d}, \frac{i}{d})$. We arbitrarily fix $d$ to be 16 in this paper, observing no significant difference when changing it to 8 and 32 in preliminary experiments. All $w_{ki}$ are enforced to be non-negative by always using the absolute value of them, which yields the monotonicity property. Dividing the weights by $\sum_m^d |w_{km}|$ yields the property that $f(1) = 1$. The function can be written as

$$f_k(x) = \sum_{i=1}^{d} \max(0, 1 - |dx - i|) \frac{\sum_{j=1}^{i} |w_{kj}|}{\sum_{m=1}^{d} |w_{km}|} \tag{12}$$

In essence, the $\max$ term selects the two nearest boundary values of an interval, which are normalized cumulative sums over the $w_k$ weights, and linearly interpolates between the two. This approach is similar to the subgradient approach by Jaderberg et al. (2015) to make sampling from indices differentiable. All $w_{ki}$ are initialized to 1, which makes the functions linear on initialization. When applying $f_k(\mathbf{x})$ to a vector-valued input $\mathbf{x}$, it is assumed to be applied elementwise. By caching the normalized cumulative sum $\sum_j^i |w_{kj}|/\sum_m^d |w_{km}|$, this function has linear time complexity with respect to $d$ and is efficiently implementable on GPUs.

Extensions to this are possible through Deep Lattice Networks (You et al., 2017), which preserve monotonicity across several nonlinear neural network layers. They would allow $\mathbf{A}$ and $\mathbf{D}$ to be combined in more sophisticated ways beyond an elementwise product, possibly improving counting performance as long as the property of the range lying within $[0, 1]$ is still enforced in some way.

## B    BASELINE ARCHITECTURE

This model is based on the work of Kazemi & Elqursh (2017), who outperformed most previous VQA models on the VQA v1 dataset with a simple baseline architecture. We adapt the model to the VQA v2 dataset and make various tweaks that improve validation accuracy slightly. The architecture is illustrated in Figure 6. Details not mentioned here can be assumed to be the same as in their paper.

The most significant change that we make is the use of object proposal features by Anderson et al. (2017) as previously mentioned. The following tweaks were made without considering the performance impact on the counting component; only the validation accuracy of the baseline was optimized.

To fuse vision features $\mathbf{x}$ and question features $\mathbf{y}$, the baseline concatenates and linearly projects them, followed by a ReLU activation. This is equivalent to $\mathrm{ReLU}(\mathbf{W}_x\mathbf{x} + \mathbf{W}_y\mathbf{y})$. We include an additional term that measures how different the projected $\mathbf{x}$ is from the projected $\mathbf{y}$, changing the fusion mechanism to $\mathbf{x} \diamond \mathbf{y} = \mathrm{ReLU}(\mathbf{W}_x\mathbf{x} + \mathbf{W}_y\mathbf{y}) - (\mathbf{W}_x\mathbf{x} - \mathbf{W}_y\mathbf{y})^2$.

The LSTM (Hochreiter & Schmidhuber, 1997) for question encoding is replaced with a GRU (Cho et al., 2014) with the same hidden size with dynamic per-example unrolling instead of a fixed 14 words per question. We apply batch normalization (Ioffe & Szegedy, 2015) before the last linear projection in the classifier to the 3000 classes. The learning rate is increased from 0.001 to 0.0015 and the batch size is doubled to 256. The model is trained for 100 epochs (1697 iterations per epoch to train on the training set, 2517 iterations per epoch to train on both training and validation sets) instead of 100,000 iterations, roughly in line with the doubling of dataset size when going from VQA v1 to VQA v2.

Note that this single-model baseline is regularized with dropout (Srivastava et al., 2014), while the other current top models skip this and rely on ensembling to reduce overfitting. This explains why our single-model baseline outperforms most single-model results of the state-of-the-art models. We found ensembling of the regularized baseline to provide a much smaller benefit in preliminary experiments compared to the results of ensembling unregularized networks reported in Teney et al. (2017).

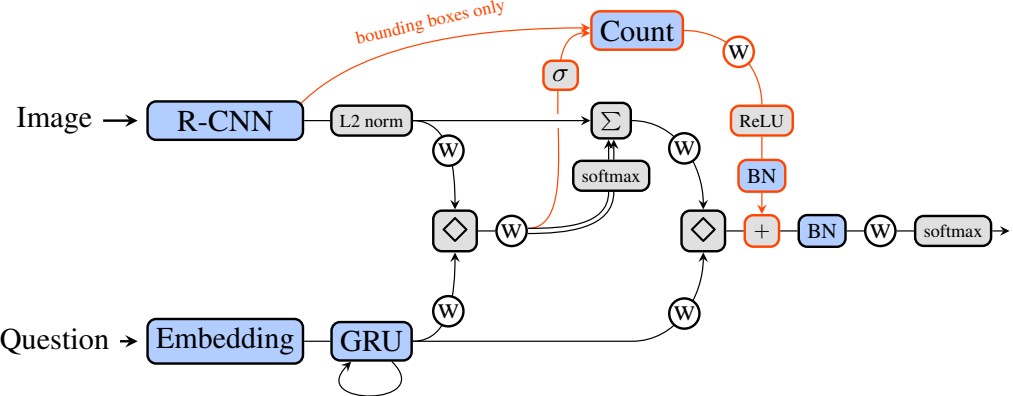

Figure 6: Schematic view of a model using our counting component. The modifications made to the baseline model when including the counting component are marked in red. Blue blocks mark components with trainable parameters, gray blocks mark components without trainable parameters. White ⓦ mark linear layers, either linear projections or convolutions with a spatial size of 1 depending on the context. Dropout with drop probability 0.5 is applied before the GRU and every ⓦ, except before the ⓦ after the counting component. ◇ stands for the fusion function we define in Appendix B, BN stands for batch normalization, $\sigma$ stands for a logistic, and Embedding is a word embedding that has been fed through a tanh function. The two glimpses of the attention mechanism are represented with the two lines exiting the ⓦ. Note that one of the two glimpses is shared with the counting component.

## C  FULL PLOTS OF ACTIVATION FUNCTIONS

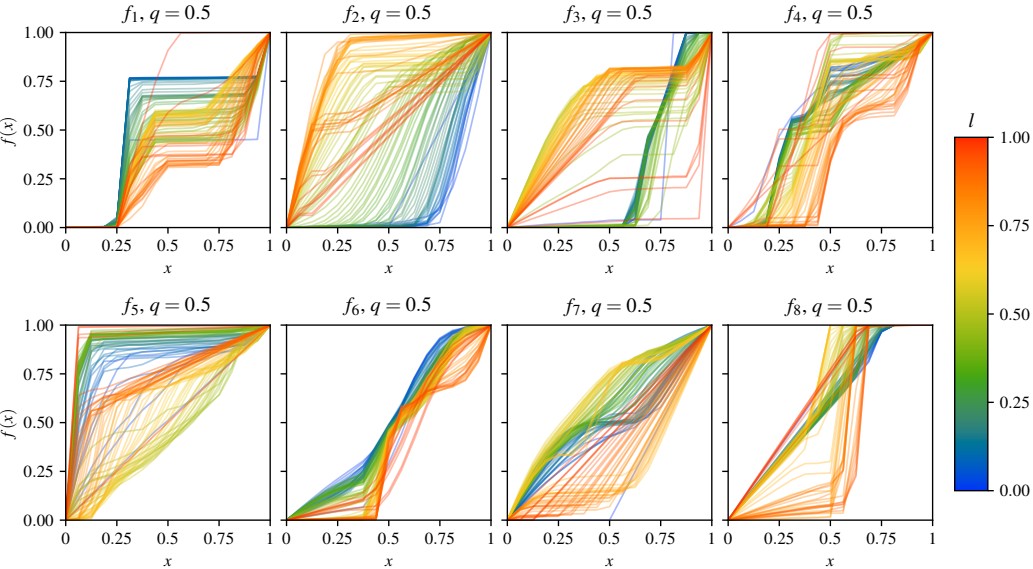

Figure 7: Shape of activation functions as $l$ is varied for $q = 0.5$ on the toy dataset. Each line shows the shape of the activation function when $l$ is set to the value associated to its color. Best viewed in color.

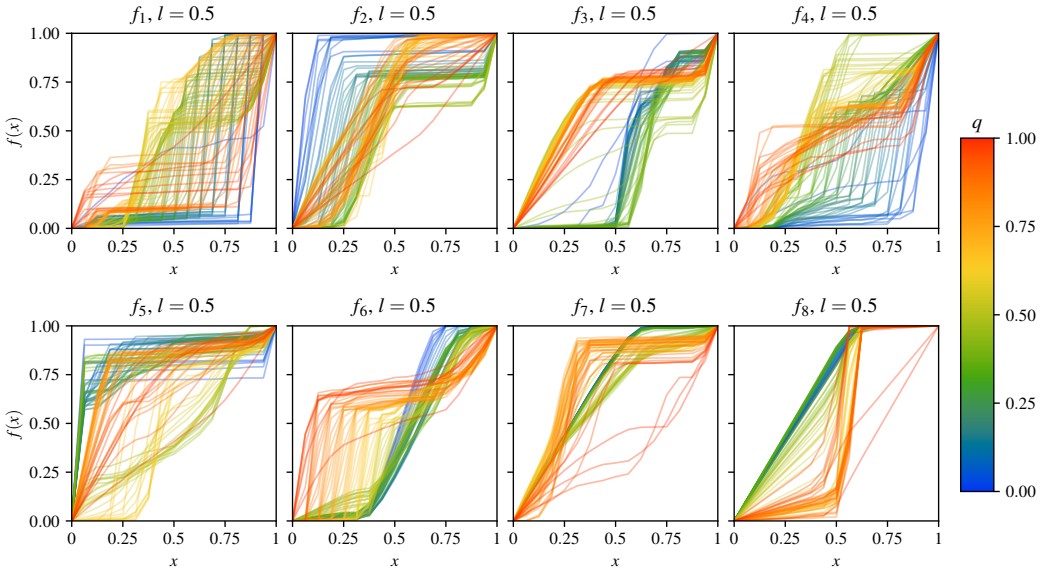

Figure 8: Shape of activation functions as $q$ is varied for $l = 0.5$ on the toy dataset. Each line shows the shape of the activation function when $q$ is set to the value associated to its color. Best viewed in color.

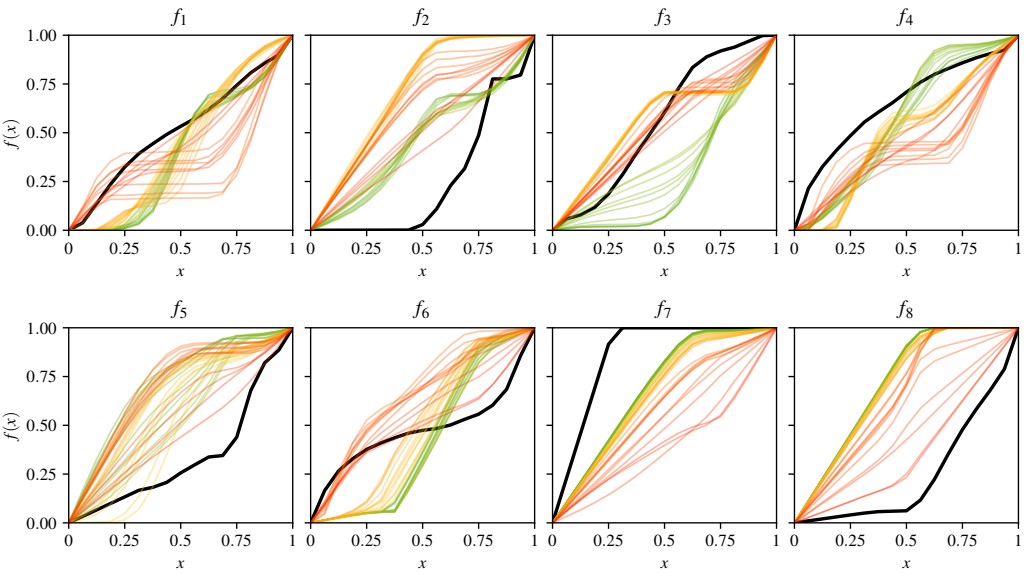

Figure 9: Shape of activation functions for a model trained on the train and validation sets of VQA v2 (thick black), compared against the shapes when parametrizing the toy dataset with $q$ around 0.4 (green), 0.7 (orange), or 1.0 (red) with fixed $l = 0.2$. Best viewed in color.

# D  EXAMPLE TOY DATASET DATA

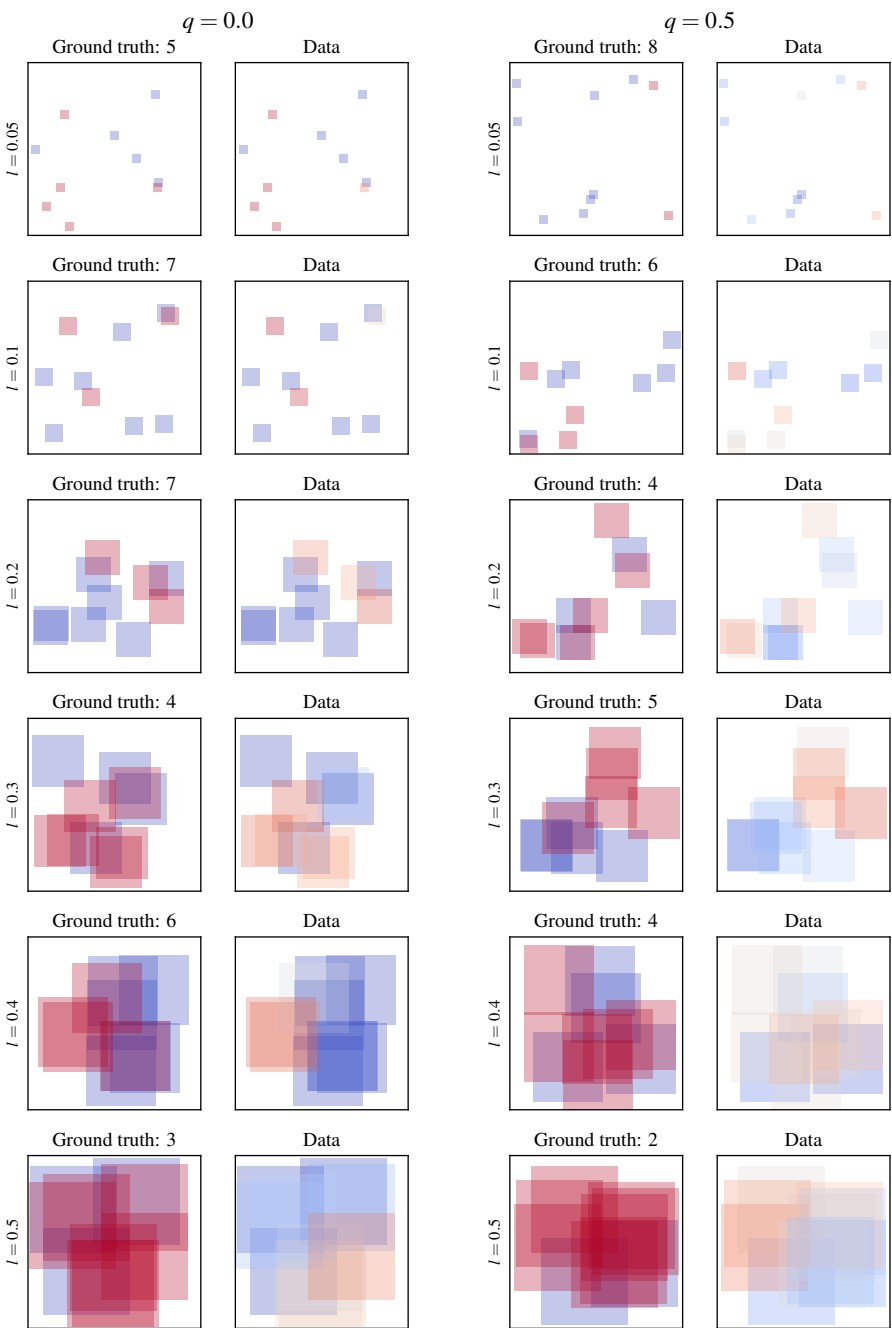

Figure 10: Example toy dataset data for varying bounding box side lengths $l$ and noise $q$. The ground truth column shows bounding boxes of randomly placed true objects (blue) and of irrelevant objects (red). The data column visualizes the samples that are actually used as input (dark blues represent weights close to 1, dark reds represent weights close to 0, lighter colors represent weights closer to 0.5). The weight of the $i$th bounding box $b_i$ is defined as $a_i = (1 - q)$ score $+ qz$ where the score is the maximum overlap of $b_i$ with any true bounding box or 0 if there are no true bounding boxes and $z$ is drawn from $U(0, 1)$. Note how this turns red bounding boxes that overlap a lot with a blue bounding box in the ground truth column into a blue bounding box in the data column, which simulates the duplicate proposal that we have to deal with. Best viewed in color.

## E    QUALITATIVE EXAMPLES OF INTERMEDIATE ACTIVATIONS

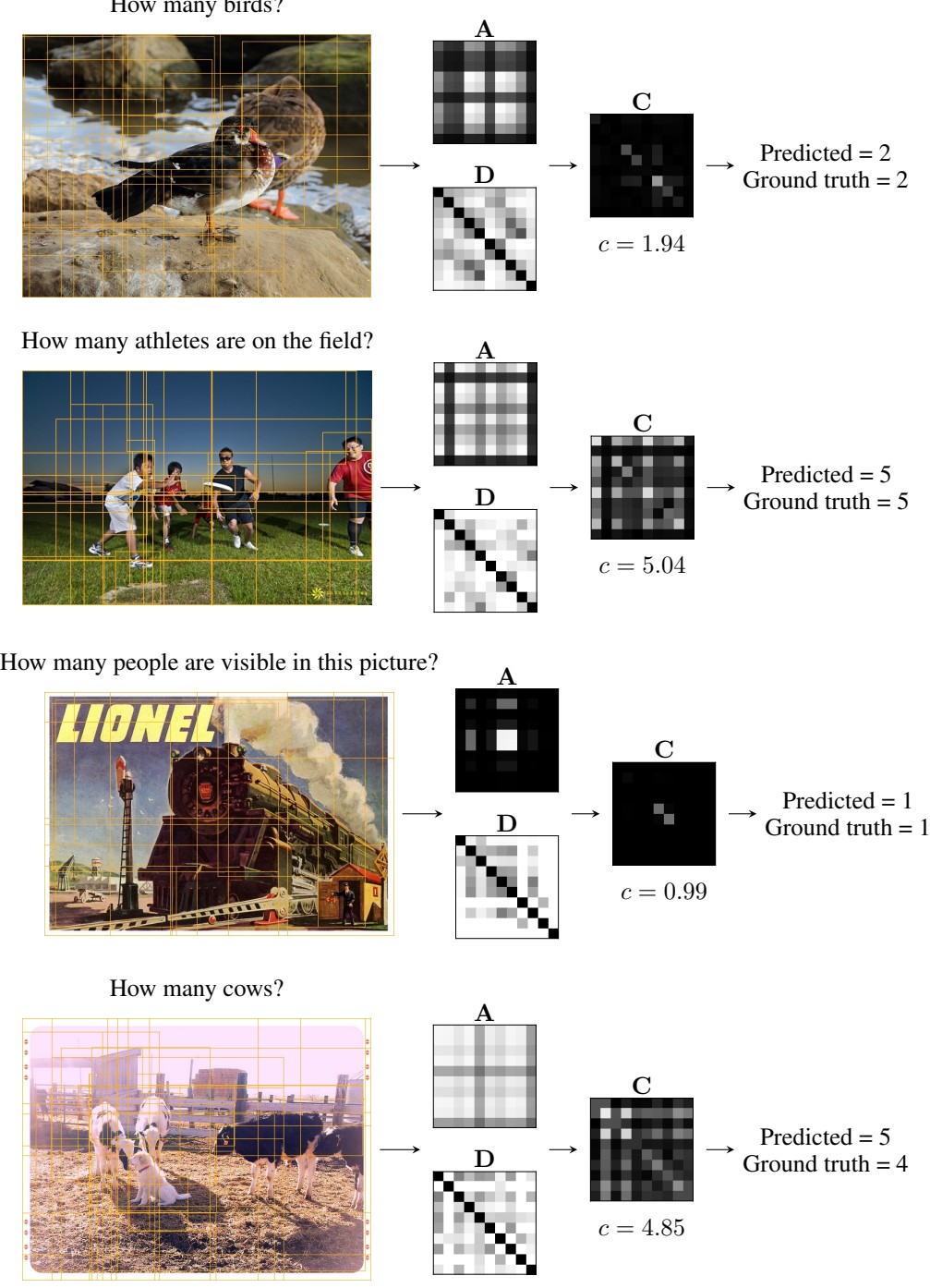

Figure 11:   Selection of validation images with overlaid bounding boxes, values of the attention matrix $\mathbf{A}$, distance matrix $\mathbf{D}$, and the resulting count matrix $\mathbf{C}$. White entries represent values close to 1, black entries represent values close to 0. The count $c$ is the usual square root of the sum over the elements of $\mathbf{C}$. Notice how particularly in the third example, $\mathbf{A}$ clearly contains more rows/columns with high activations than there are actual objects (a sign of overlapping bounding boxes) and the counting module successfully removes intra- and inter-object edges to arrive at the correct prediction regardless. The prediction is not necessarily – though often is – the rounded value of $c$.

