# OpenReview forum: "Learning to Count Objects in Natural Images for Visual Question Answering"
_ICLR.cc/2018/Conference — Accept (Poster)_

### Official Review · AnonReviewer1 · 2017-11-27
**Counting in VQA**

**Rating:** 6
**Confidence:** 3

**Review:**

Summary
 - This paper mainly focuses on a counting problem in visual question answering (VQA) using attention mechanism. The authors propose a differentiable counting component, which explicitly counts the number of objects. Given attention weights and corresponding proposals, the model deduplicates overlapping proposals by eliminating intra-object edges and inter-object edges using graph representation for proposals. In experiments, the effectiveness of proposed model is clearly shown in counting questions on both a synthetic toy dataset and the widely used VQA v2 dataset.

Strengths
 - The proposed model begins with reasonable motivation and shows its effectiveness in experiments clearly.
 - The architecture of the proposed model looks natural and all components seem to have clear contribution to the model.
 - The proposed model can be easily applied to any VQA model using soft attention.
 - The paper is well written and the contribution is clear.

Weaknesses
 - Although the proposed model is helpful to model counting information in VQA, it fails to show improvement with respect to a couple of important baselines: prediction from image representation only and from the combination of image representation and attention weights.
 - Qualitative examples of intermediate values in counting component--adjacency matrix (A), distance matrix (D) and count matrix (C)--need to be presented to show the contribution of each part, especially in the real examples that are not compatible with the strong assumptions in modeling counting component.

Comments
 - It is not clear if the value of count "c" is same with the final answer in counting questions.

---

> ### Author Response · Authors · 2017-12-06
> **Rebuttal**
>
> Thank you for the review. We are glad to hear that you think that everything is reasonably motivated, the results are good, and that there is a clear contribution with good writing.
>
>
> -    Fails to show improvement with respect to a couple of important baselines
>
> Can you elaborate on what you mean with "image representation only" and "combination of image representation and attention weights"? We are not sure whether you are referring to existing experiments in the paper or experiments that you would like to see (we are happy to include these baselines if reasonable). Just to clarify the existing baselines that we already compare against: all the models in Table 1 and Table 2 use soft attention with softmax normalization on object proposal features. We did not list models using pixel representations, since they are outperformed by models using object proposals in all question categories of VQA (Andersen et al., 2017). Models with attention have been shown to outperform models without attention many times in the literature (e.g. survey in [1]).
>
>
> -    Qualitative examples of matrices A, D, and C are needed
>
> Thank you for the good idea. We will include some examples of these in a revision of the paper.
>
>
> -    Unclear whether c is the same as the predicted answer
>
> c is not necessarily the predicted answer; it is just a feature (which gets turned into a linear interpolation of appropriate one-hot encoded vectors as per equation 8) that the answer classifier makes use of. Since not all questions in VQA are counting questions, the model learns how and when to use this feature. The existing model descriptions in 5.1 and 5.2, along with the diagram of the VQA model architecture, should make this clear.
>
>
> Additional references:
> [1] Damien Teney, Qi Wu, and Anton van den Hengel. Visual Question Answering: A Tutorial. In IEEE Signal Processing Magazine, vol. 34, no. 6, pp. 63-75. http://ieeexplore.ieee.org/stamp/stamp.jsp?tp=&arnumber=8103161&isnumber=8103076

---

### Official Review · AnonReviewer2 · 2017-11-27
**Improve object counting with a lot of heuristics**

**Rating:** 4
**Confidence:** 4

**Review:**

This paper tackles the object counting problem in visual question answering. It is based on the two-stage method that object proposals are generated from the first stage with attention. It proposes many heuristics to use the object feature and attention weights to find the correct count.  In general, it treats all object proposals as nodes on the graph. With various agreement measures, it removes or merges edges and count the final nodes. The method is evaluated on one synthetic toy dataset and one VQA v2 benchmark dataset. The experimental results on counting are promising.  Although counting is important in VQA, the method is solving a very specific problem which cannot be generalized to other representation learning problems.  Additionally, this method is built on a series of heuristics without sound theoretically justification, and these heuristics cannot be easily adapted to other machine learning applications. I thus believe the overall contribution is not sufficient for ICLR.

Pros:
1. Well written paper with clear presentation of the method.
2. Useful for object counting problem.
3. Experimental performance is convincing.

Cons:
1. The application range of the method is very limited.
2. The technique is built on a lot of heuristics without theoretical consideration.

Other comments and questions:

1. The determinantal point processes [1] should be able to help with the correct counting the objects with proper construction of the similarity kernel.  It may also lead to simpler solutions. For example, it can be used for deduplication using A (eq 1) as the similarity matrix.

2. Can the author provide analysis on scalability the proposed method? When the number of objects is very large, the graph could be huge. What are the memory requirements and computational complexity of the proposed method?
In the end of section 3, it mentioned that "without normalization," the method will not scale to an arbitrary number of objects. I think that it will only be a problem for extremely large numbers. I wonder whether the proposed method scales.

3. Could the authors provide more insights on why the structured attention (etc) did not significantly improve the result? Theoritically, it solves the soft attention problems.

4. The definition of output confidence (section 4.3.1) needs more motivation and theoretical justification.

[1] Kulesza, Alex, and Ben Taskar. "Determinantal point processes for machine learning." Foundations and Trends® in Machine Learning 5.2–3 (2012): 123-286.

---

> ### Author Response · Authors · 2017-12-06
> **Rebuttal Part 1/3: Applications beyond VQA**
>
> Thank you for your review. We are glad to see that you think the paper is well written and that the evidence for the usefulness to object counting is convincing. Given this, we are slightly surprised by the low rating as we think that the pros you list are very concrete compared to the cons.
> In summary, we disagree with the main claims that the application range is very limited and that it is built on heuristics without theoretical considerations.
>
>
> -    Method is solving a very specific problem which cannot be generalized to other representation learning problems / application range of the method is very limited
>
> We disagree that the application of this method is limited to VQA. In fact, the toy task that we perform experiments on is clearly not a VQA task and shows that the component is applicable beyond VQA. Counting tasks have many practical real-world applications on their own.
>
> An immediate research area wherein it can be used aside from VQA is image caption generation, where an attention LSTM can be used to attend over objects (Anderson et al, 2017). In this task, counting information should be useful for generating good captions and the attention mechanism has the same limitations as we discuss in section 3, which our counting component can be used for. Any task where counting of specific objects (without necessarily conditioning on a question input) is required but no ground truth bounding boxes are available -- which limits the use of some conventional methods for training counting models -- can use a pre-trained region proposal network. The score of a binary classification on each proposal whether it shows the object to count can be used as attention weight in the component to eliminate duplicates without the need for post-processing with non-maximum suppression and score thresholding, both of which require hyper-parameter tuning and disallow end-to-end training. This system can be trained end-to-end with the counting module, allowing a sensible approach for handling duplicate and overlapping bounding boxes.
>
> More generally, tasks where a set of potential objects (wherein each object is given a score of how relevant it is) with possible duplicates needs to be counted, and duplicates can be identified through a pairwise distance metric (in the image case these are the 1 - IoU distances of bounding boxes), the component can be used for counting the objects with duplicates eliminated in a fully differentiable manner. Most importantly, appropriate relevancy scores and distances do not need to be specified explicitly as they can be learnt from data.
>
> As we have shown in the paper, the component is robust enough to count without per-object ground-truth as supervision; only the aggregate count is needed. This makes it applicable to a wide variety of counting tasks beyond VQA.
>
>
> -    Heuristics cannot be easily adapted to other machine learning applications
>
> While the properties that we use are specifically targeted towards counting, we think that there is value to be gained for the wider research community from our general approach to the problem. Our insight about the necessity of using the attention map itself, not just the feature vector coming out of the attention, may lead to recognition of problems that soft attention can introduce in other domains such as NLP, which in turn can lead to new solutions. The approach of a learned interpolation between correct behaviours, enforced by the network structure through monotonicity, may also be useful. For the monotonicity property, networks with more nonlinearities such as Deep Lattice Networks (You et al., 2017) can be used as well as we mention in Appendix A. Our way of treating the counting problem as a graphical problem and decomposing it into intra- and inter-object relations may find use in problems where there is some notion of object under uncertainty. The approach of creating a fully differentiable model despite many operations naively being non-differentiable, in particular when we want to remove certain edges but instead use equivalence under a sum to our benefit, contributes to the growing literature (e.g. (Jaderberg et al., 2015)) of making operations required for certain tasks differentiable and thus trainable in a deep neural network setting.

---

> > ### Author Response · Authors · 2017-12-06
> > **Rebuttal Part 2/3: Theory, determinantal point processes and scalability**
> >
> > -    Built on a lot of heuristics without theoretical consideration
> >
> > We disagree that there is no theoretical consideration in our method. Mathematically, we are handling the edge cases for counting correctly and all other behaviour is based on a learned interpolation between the edge cases, guaranteeing at least some sort of sensible counting. It is true that it is not based on a well-established mathematical framework, but we are successfully solving a practical problem with a practical solution. Every step within the component is theoretically justified with what property is necessary in order for the component to produce correct counts, followed by a way of achieving each property. After building up the model this way, it produces perfect counts in the cases when our modeling assumptions hold (the properties that we claimed are necessary lead to this) and the performance degrades gracefully under uncertainties when our modeling assumptions do not hold (another consequence of the properties that we enforce). Thus, we disagree with your claim that these are theoretically unmotivated heuristics; every part of the component has a reasonable, theoretically-based motivation based on what a correct counting mechanism should do. AnonReviewer1 agrees that all steps are reasonably motivated in their review.
> >
> > We do not think that this paper should be rejected simply because it is does not use an established mathematical model to solve a task. This complaint seems to apply to much of Deep Learning and Computer Vision research in general, not this paper in particular.
> >
> >
> > -    Determinantal point process should be able to help
> >
> > This connection looks interesting, but from the survey it is not clear that it is at all applicable to counting. The mathematics may be well grounded, but the assumptions about diversity seem very ad-hoc. Thus, we think that using DPPs for counting would be an unjustified heuristic. Our approach may not be grounded in a mathematical formulation, but we are correctly handling edge cases and allowing a suitable interpolation to be learned from data. To our knowledge, DPPs have found very little use in the field of Deep Learning so far. We are only aware of [1] which uses DPPs for model compression, which is evidently unrelated to the task of counting.
> >
> >
> > -    Scalability of the proposed method
> >
> > As stated in section 4.2.2, the time complexity is Theta(n^3) where n is the number of objects used. This can be reduced using the alternative similarity measure that we mention before that to Theta(n^2), though all results reported use the former similarity. The space complexity is Theta(n^2), as the matrices A, D, and C each have n^2 elements. Here are some numbers of our implementation, showing approximate time taken for one training epoch of the whole model on VQA v2 and amount of memory allocated on a Titan X (Pascal) GPU. The times for the low max object counts are very rough and are averaged across a few epochs -- as training time typically changes by about 10 seconds epoch by epoch -- and memory usage can vary slightly between runs.
> >
> > max objects, time (minutes), memory (MiB)
> > 1, 5:50, 3701
> > 10 (default), 6:00, 3715
> > 25, 6:15, 4095
> > 50, 9:50, 7393
> > 60, 12:50, 10779
> >
> > As you can see, increasing the number of objects from 10 to 25 incurs only small additional computational costs and even going to 50 objects, operating on a 2500 entry matrix per example, is still quite reasonable. For practical cases that we are dealing with in VQA, where we limit the maximum number of objects to 10 -- this covers the vast majority of counting questions -- using the counting component uses marginal amounts of extra computational resources. We do not claim that the method will be applicable to huge graphs and it is probably not the best mechanism for counting a large number of objects. There are also several ways of reducing the run time for large numbers of objects through e.g. a k-d tree with the reasonable assumption that when there are many objects to count, an object does not overlap with all other objects. The main difficulty of VQA is that most of the time the number of objects to count is relatively small; in contrast, the queries of what objects to count and the spatial relationships between objects can be complex.

---

> > > ### Author Response · Authors · 2017-12-06
> > > **Rebuttal Part 3/3: Issues of scale with existing methods, structured attention, and output confidence**
> > >
> > > -    Claim of "without norm" the method doesn't scale to arbitrary numbers.
> > >
> > > To keep things clear: we are talking about methods that apply unnormalized attention weights to a set of feature vectors and trying to count from the resulting single feature vector (or multiple if there are multiple attention glimpses). When referring to not being able to scale to arbitrary numbers, we are referring to numbers even beyond 2. Using this method, scaling the whole feature vector should have the effect of scaling the count to predict, since that is exactly what happens when the attention weights are not normalized and the input is simply duplicated (as per the example in section 3). The issue is that the model has to learn that joint scaling of all features (2048 features per glimpse in our case) is related to count, but scaling of individual features is still related to different levels of expression of that feature in the input. These two properties seem contradictory to us; when feature vectors in the input set can vary from each other greatly, it is unclear to us how the joint scaling of all features property can be learnt at all beyond tiny changes in scale. It also contradicts the common notion of being able to approximately linearly interpolate in the latent space of deep neural networks, since the magnitude of a feature is no longer directly related to the expression of that feature, but depends on the magnitude of all other features in a fairly complex relationship. Empirically, using a sigmoid activation or simple averaging across the set of feature vectors without attention has not helped in several previous works, neither for counting nor for overall performance as mentioned at the end of section 3.
> > >
> > > Thus, we highly doubt that sigmoid activation or similar methods that do not normalize attention weights to sum to 1, despite leaving more information about counting in the feature vector than softmax normalization, can lead to feature vectors from which counting can be learned at all. As we discuss in section 2, any improvement in counting you see despite this requires counting information to already be present in the input, which limits the objects that can be counted to things that the object proposal network can distinguish. When saying that it does not scale to arbitrary numbers, we were conservative in that statement in that we can imagine that in special cases it might be possible to learn to relate very small joint feature scaling to counting information, but not generally or in practice.
> > >
> > > We realize that this is a slightly alternative explanation than we provide in the paper and will update the paper to make this clearer accordingly.
> > >
> > > -    Insights on why structured attention did not significantly improve the result
> > >
> > > With structured attention, each individual glimpse can select a contiguous region of pixels associated to individual objects unlike regular soft attention. However, it still lacks a way of extracting information from the attention map itself, which is necessary for counting as we argue in section 3. In order to attend to multiple objects, multiple glimpses are needed as well. This makes structured attention on pixels very similar to soft attention on object proposals; the structure in the attention acts as an implicit object detector. Thus, while structured attention solves one problem with soft attention -- the same that using object proposals solves -- it is not enough to actually count.
> > >
> > >
> > > -    Output confidence needs more motivation and theoretical justification
> > >
> > > We think that we have provided sufficient motivation for the output confidence in the paper.
> > > Here is an expanded version of the consequences of it: The output confidence can learn to suppress the magnitude of the counting features on an example-by-example basis by how close the values of vector a and matrix D are to the ideal values. When for certain values of D and a the predicted count is inaccurate during training, the gradient update reduces the magnitude of the counting features for those values through the output confidence. This lets the counting component learn when it is inaccurate and allows the VQA model using the component to compensate for it instead of blindly trusting that the counting features are always reliable. We found this to be a useful -- though not absolutely necessary -- step that slightly improves counting performance in practice.
> > >
> > >
> > > Additional references:
> > > [1] Zelda Mariet and Suvrit Sra. Diversity Networks: Neural Network Compression using Determinantal Point Processes. In ICLR, 2016.

---

### Official Review · AnonReviewer3 · 2017-11-28
**Model is too hand-crafted and key experiments missing**

**Rating:** 6
**Confidence:** 3

**Review:**


Summary:
- This paper proposes a hand-designed network architecture on a graph of object proposals to perform soft non-maximum suppression to get object count.

Contribution:
- This paper proposes a new object counting module which operates on a graph of object proposals.

Clarity:
- The paper is well written and clarity is good. Figure 2 & 3 helps the readers understand the core algorithm.

Pros:
- De-duplication modules of inter and intra object edges are interesting.
- The proposed method improves the baseline by 5% on counting questions.

Cons:
- The proposed model is pretty hand-crafted. I would recommend the authors to use something more general, like graph convolutional neural networks (Kipf & Welling, 2017) or graph gated neural networks (Li et al., 2016).
- One major bottleneck of the model is that the proposals are not jointly finetuned. So if the proposals are missing a single object, this cannot really be counted. In short, if the proposals don’t have 100% recall, then the model is then trained with a biased loss function which asks it to count all the objects even if some are already missing from the proposals. The paper didn’t study what is the recall of the proposals and how sensitive the threshold is.
- The paper doesn’t study a simple baseline that just does NMS on the proposal domain.
- The paper doesn’t compare experiment numbers with (Chattopadhyay et al., 2017).
- The proposed algorithm doesn’t handle symmetry breaking when two edges are equally confident (in 4.2.2 it basically scales down both edges). This is similar to a density map approach and the problem is that the model doesn’t develop a notion of instance.
- Compared to (Zhou et al., 2017), the proposed model does not improve much on the counting questions.
- Since the authors have mentioned in the related work, it would also be more convincing if they show experimental results on CL

Conclusion:
- I feel that the motivation is good, but the proposed model is too hand-crafted. Also, key experiments are missing: 1) NMS baseline 2) Comparison with VQA counting work  (Chattopadhyay et al., 2017). Therefore I recommend reject.

References:
- Kipf, T.N., Welling, M., Semi-Supervised Classification with Graph Convolutional Networks. ICLR 2017.
- Li, Y., Tarlow, D., Brockschmidt, M., Zemel, R. Gated Graph Sequence Neural Networks. ICLR 2016.

Update:
Thank you for the rebuttal. The paper is revised and I saw NMS baseline is added. I understood the reason not to compare with certain related work. The rebuttal is convincing and I decided to increase my rating, because adding the proposed counting module achieve 5% increase in counting accuracy. However, I am a little worried that the proposed model may be hard to reproduce due to its complexity and therefore choose to give a 6.

---

> ### Author Response · Authors · 2017-12-06
> **Rebuttal Part 1/4: Hand-crafted nature**
>
> Thank you for your review. We are happy to see that you think the paper is well written and that the deduplication steps in the module are interesting. Given the aptness of your comments, it seems like you understand the paper better than you are giving yourself credit for.
> In summary to your main complaints: We argue that a hand-crafted approach is reasonable for the current state of VQA, NMS baselines will be supplied, and comparisons to (Chattopadhyay et al., 2017) are not useful.
>
> -    Proposed model is pretty hand-crafted, would recommend the authors to use something more general, like graph convolutional neural networks.
>
> In summary, we think that with current VQA models, counting needs to be hand-crafted to some extent, hand-crafting counting has various useful properties, and that we tried a non-handcrafted approach similar to graph convolutional networks in the past without success.
>
> We think that with the current state of VQA models on real images, it is unreasonable to expect a general model to learn to count without hand-designing some aspect of it in order for the model to learn. Pointed out many times such as in (Jabri et al., 2016), [1], and seen from the balanced pair accuracies in Table 2, much of current VQA performance is due to fitting better to spurious dataset biases with little "actual" learning of how to answer the questions. The necessity for a modularized approach is also recognized in a recently published work in NIPS [2], where they combine a variety of different types of models, each suited to a different pre-defined task (e.g. one face detection network, one scene classification network, etc.). The aspect that makes the counting task within VQA special is that there is some relatively easy-to-isolate logic to it, which is the focus of our module through soft deduplication and aggregation. Even in humans, counting is a highly structured process when going beyond the range where humans can subitize. While it would certainly be better if a neural network could discover the logic required for counting by itself, we think that a hand-engineered approach is perfectly valid for solving this problem given the current state of research and performance on VQA.
>
> The hand-crafted nature gives the component several useful properties. Due to the structure of the component, it performs more-or-less correct counting by default, even when none of the parameters in it have been learnt yet. This allows it to generalize more easily to a test set with fewer training samples and under much noise, as is the case for VQA. Since all steps within the component have a clear motivation, the parameters that it learns are interpretable and can be used for explaining why it predicted a certain count. Changing the input has a predictable effect on the output due to the component structure enforcing monotonicity. This is particularly useful in comparison to a general deep neural network, which suffers from adversarial inputs causing unexpected predictions. The simple nature of the module with relatively few parameters keeps the computational costs low and allows it to be integrated into non-VQA tasks fairly easily. Note that the modeling assumptions that we make are not specific to VQA, but are assumptions about what a sensible counting method should do in ideal cases.
>
> In our experience, integrating other types of models into VQA models is difficult without either inhibiting general performance or simply achieving essentially the same level of performance. As far as we are aware, there has not been any work which successfully uses a graph-based approach to VQA on real images. We did try to integrate relation networks (Santoro et al., 2017) into a VQA model, without much success in terms of performance on counting nor in any other category (though this obviously does not mean that a successful integration is not possible). Relation networks are a natural choice for VQA v2, perhaps more so than the neural networks for arbitrary graphs you suggest: they have been shown to work well for VQA on the CLEVR dataset and treat objects as nodes in a complete graph, similar to what our module uses as input. With our module, we at least show that a graph-based representation can find some use in VQA on real images in the first place and might motivate further research into graph-based approaches. In general, the sorts of graph-based approaches that you mention have only been successfully applied on the abstract VQA dataset so far [3], where a precise scene graph of synthetic images is used as input, not real images. On that dataset, good improvements in counting have been achieved by a general graph-based network. We imagine that this is due to the much less noisy nature of scene graphs on synthetic data compared to using pixel-based representations or object proposals on real images, making counting a much easier task in the abstract VQA case.

---

> > ### Author Response · Authors · 2017-12-06
> > **Rebuttal Part 2/4: Joint fine-tuning of proposals and NMS baseline**
> >
> > -    Proposals are not jointly finetuned, did not study recall of the proposals and how sensitive the threshold is
> >
> > Can you clarify what threshold you are referring to in your comment? There is no hard threshold anywhere in our model.
> >
> > This is certainly a valid concern, but applies to all VQA models using object proposals, not just our counting module. If the loss for counting is biased, then so is the loss for the rest of the model (e.g. "what color is the car" without an object proposal on the car). Joint training is nontrivial, since the architecture that generates the object proposal bounding boxes and features (Faster R-CNN) uses a two stage approach for training anyway and requires ground truth bounding boxes of objects. It is not clear to us whether joint training is at all possible in VQA, since it does not have ground truth bounding boxes available. Empirically, we are getting a substantial improvement in counting performance, so we think that the lack of joint training is not a major issue. This issue is certainly something that can be looked into more in the future, but we do not see it as a shortcoming of our module in particular.
> >
> > While we do not think that it is our responsibility to evaluate proposal recall, we are looking into manually labeling a small subset of training examples to get a sense of how much of an issue this is in general. One thing to keep in mind is that the loss pushes the VQA model as a whole to predict a certain answer, not just the counting component itself. That means that a bias towards not recognizing some types of objects (either in the object proposal network or the attention mechanism) can be accounted for by the rest of the model by biasing the count predictions of these types of objects slightly upwards. Bounding boxes that capture different parts of one object (e.g. one capturing the upper half of a person and one capturing the lower half of the same person) can also still lead to a correct count prediction if the attention mechanism recognizes that half the attention weight as usual should be given to those. In general, it might be enough for the counting module to produce a sensible prediction as long as some number of bounding boxes cover the all the required objects, not necessarily with one box per object.
> >
> >
> > -    Doesn't study a simple baseline that just does NMS on the proposal domain
> >
> > Thank you for pointing out the lack of this baseline. We agree that this should have been included and we have started running experiments for this. Initial experiments are suggesting that when the counting module is replaced with the one-hot encoded number of objects determined by NMS (we are trying thresholds between 0.3 and 0.7) the performance is not much different, if at all, from the baseline without NMS. This applies to using one of the two attention maps (like the counting module) as well as the sum of the two attention maps (lack of gradient means that the model can't specialize the first attention map to locate the objects to count, so using the sum of the two attention maps might be more reasonable) for scoring the proposals, which suggests that the piecewise constant gradient of NMS is a major issue. Once we have the full results, we will certainly include this information in a revision to the paper.

---

> > > ### Author Response · Authors · 2017-12-06
> > > **Rebuttal Part 3/4: Comparison with Chattopadhyay and symmetry breaking**
> > >
> > > -    Doesn't compare experiment numbers with (Chattopadhyay et al., 2017)
> > >
> > > There are several major differences that make a direct comparison to the results in their work not useful (we have confirmed these differences with Chattopadhyay).
> > >
> > > 1. They create a subset of the counting question subset of VQA v1, but their model is not trained on it. It is trained on the ~80 000 COCO training images with a ground truth labeling of how many objects there are for each of the 80 COCO classes, in essence giving them ~6 400 000 counts to train with. In contrast, there are only ~50 000 counting questions in the training set of VQA v2 (which is around twice the size of VQA v1), with the added difficulty of the types of objects being arbitrarily complex (e.g. "how many people" vs "how many people wearing brown hats").
> > > 2. When they evaluate their model on VQA, they select a small subset (roughly 10%--20% of the counting question subset in VQA v1) where the ground-truth count of the COCO class that their NLP processing method extracts from the question is the same as the VQA ground-truth count. During evaluation, they run their method on the input image as usual, and simply use the output corresponding to the extracted class as prediction. This means that they are essentially evaluating on a subset of the COCO data that they previously evaluated on already, or conversely, only using the subset of VQA that basically matches the COCO validation data anyway. We feel that it is a stretch to call this a VQA task, since at no point any VQA is actually performed in their model.
> > > 3. The VQA models are solving a slightly different task: unlike their proposed models, the VQA models are processing a natural language question, which may go wrong for the VQA models but is ensured to be correct for their proposed models (since they discard any examples where their NLP processing scheme gets it wrong). Additionally, VQA models are trained to not only try to answer counting questions, but also other questions.
> > >
> > > Due to these disadvantages to regular VQA models in their setup, we doubt that the performance of their model can be adequately compared to ours. In order for a comparison to be useful, we would have to train with the same training data that they used, which we feel is too much of a departure from the VQA setting in this paper; the general structure of the models they use don't have much to do with VQA models in the first place (their models regress counts for the 80 COCO classes simultaneously, whereas VQA models have an additional input -- the question -- which determines what to count and then classify a count answer).
> > >
> > > We agree that superficially their results look related and we will clarify this matter in our next paper revision.
> > >
> > >
> > > -    Doesn't handle symmetry breaking
> > >
> > > We think that when the goal is to count, it is better for counting performance to not break symmetries, without having the limitation of producing discrete instances. For example, consider the case where there are 4 non-overlapping objects, each with a weight of 1/2. All edges have the same weight, but it is not clear whether there is a sensible way as to how the symmetry should be broken here. There is much precedence in Machine Learning for this type of approach, e.g. in a mixture of 2 Gaussians model, a sample in-between the two distributions is assigned to each distribution with an equal weight, rather than having a hard assignment of this sample to one distribution or the other.
> > >
> > > We agree that having instances has clear benefits over the density map style approach in terms of interpretability. However, we don't think that current attention models are good enough yet, i.e. consistently produce scores either very close to 0 or 1, to be able for an approach with instances to be as accurate as without. Thus, we think that the density-map-like approach is appropriate for counting and not a problem.

---

> > > > ### Author Response · Authors · 2017-12-06
> > > > **Rebuttal Part 4/4: Comparisons to Zhou and CLEVR**
> > > >
> > > > -    Not much improvement compared to (Zhou et al., 2017)
> > > >
> > > > This is not a like-to-like comparison. Note that their model is an ensemble of 8 models wherein each individual model already performs significantly better than our baseline without counting module, due to the use of their state-of-the-art multimodal pooling method and pre-trained word embeddings. To be precise, their single model has a better overall accuracy by about 1.3%, which widens to a difference of about 3.2% after ensembling (we have only recently obtained their single-model results and will update the paper accordingly to make this clearer). Their single-model also exploits the existing primitive features better and starts with 2.7% better accuracy in number questions (these are the primitive counting features we discuss in section 2). Despite this difference in starting performance, our relatively simple baseline without their elaborate multimodal fusion outperforms their single model by over 2% and even their ensemble by about 0.3% in the number category, just by including the counting component in the model and without ensembling our model. Since their method should improve the quality of attention maps, we expect the benefit of the counting module -- which relies on the quality of attention maps -- to stack with their improvements. Keep in mind that their soft attention uses regular softmax normalization, which means that the limitations with respect to counting that we point out in section 3 apply to their model. We emphasize that the main comparison in Table 1 to make is: the performance on the number category of the baseline with counting module improves substantially compared to the baseline without the counting module and is also the best-ever reported accuracy on number questions. This shows that the more detailed results in Table 2 on the validation set are not simply due to hill-climbing on the validation set, since the test set of VQA v2 in Table 1 is only allowed to be evaluated on at most 5 times in total.
> > > >
> > > >
> > > > -    More convincing with results on CL
> > > >
> > > > More results are almost always more convincing, but we feel like there is not much value to be gained by additionally evaluating on CLEVR (assuming that you mean CLEVR with CL) and there is a limited amount of experiments that we can put in a paper. This is mainly due to our use of bounding boxes -- non-standard for this dataset and thus making comparisons to existing work less useful -- and our focus on being able to count in the difficult setting demanded of by VQA v2: noisy attention maps (due to language and attention model with free-form human-posed questions) and noisy bounding boxes overlapping in complex ways (due to object proposal model on real images). These would be present in CLEVR to some extent as well, but in terms of synthetic tasks, we think that it is more useful for us to study counting behaviour on our toy dataset and in terms of VQA tasks, VQA v2 is more suitable for showing the benefits of our module than CLEVR.
> > > >
> > > > Additional references:
> > > > [1] Damien Teney, Qi Wu, and Anton van den Hengel. Visual Question Answering: A Tutorial. In IEEE Signal Processing Magazine, vol. 34, no. 6, pp. 63-75. http://ieeexplore.ieee.org/stamp/stamp.jsp?tp=&arnumber=8103161&isnumber=8103076
> > > > [2] Ilija Ilievski and Jiashi Feng. Multimodal Learning and Reasoning for Visual Question Answering. In NIPS, 2017. http://papers.nips.cc/paper/6658-multimodal-learning-and-reasoning-for-visual-question-answering.pdf
> > > > [3] Damien Teney, Lingqiao Liu, and Anton van den Hengel. Graph-Structured Representations for Visual Question Answering. In CVPR, 2017. http://openaccess.thecvf.com/content_cvpr_2017/papers/Teney_Graph-Structured_Representations_for_CVPR_2017_paper.pdf

---

> ### Author Response · Authors · 2018-01-24
> **Response to update**
>
> Thank you for your update; we are glad to hear that you found the rebuttal convincing.
>
> With regards to your comment about worries that the proposed model may be hard to reproduce due to its complexity: As we mention in a footnote in the paper, we will open-source all of our code soon. The complexity that you perceive can be boiled down to a sequence of simple tensor operations, so we think that it should be reasonably straightforward to implement in any modern Deep Learning framework. Here is the snippet of our model implementation in PyTorch (we do not rely on the dynamic computation graph feature of PyTorch), with the important bit being the forward function of the Counter class: https://gist.github.com/anonymous/669509edc32eb28cc508221de47baa43 .  We will clean this and the rest of the code up to be easier to follow before release.

---

### Author Response · Authors · 2017-12-15
**Paper revision 1**

We have updated the paper clarifying some things that reviewers maybe misunderstood and added experimental results that the reviewers wanted to see.

- Clarified that results in (Chattopadhyay et al., 2017) are not comparable at the end of section 2. (Reviewer3)
- Improved explanation of why attention with sigmoid normalization (or similar) produces feature vectors that do not lend themselves to count at all in section 3. (Reviewer2)
- Included NMS results in Table 2. (Reviewer3)
- Clarified comparisons with (Zhou et al., 2017) in section 5.2.1. (Reviewer3)
- Included some qualitative examples of matrices A, D, and C in Appendix E. (Reviewer1)
- Explicitly state how the component has use outside of VQA in section 6. (Reviewer2)

---

### Author Response · Authors · 2018-01-04
**Related submission: Interpretable Counting for Visual Question Answering**

We would like to point out a related paper that was submitted to ICLR 2018: Interpretable Counting for Visual Question Answering, https://openreview.net/forum?id=S1J2ZyZ0Z
They also tackle the problem of counting in VQA with a sequential counting method with several differences in the approach:

- They use a more-or-less generic network for sequential counting and design a specific loss, while we design a specific network component and use a generic loss.
- We use the standard full VQA dataset, whereas they create their own dataset by taking only counting questions from the VQA and Visual Genome datasets. This makes our results comparable to prior work on VQA, showing a clear benefit over existing results in counting. In total we use more questions to train with (since we are using all VQA questions, not just counting ones), but fewer counting questions (since we are not using counting questions from Visual Genome), so the impact of this difference on counting performance is unclear.
- It is unclear to us whether their method is usable within usual VQA architectures due to their loss not applying when the question is not a counting question. Our model is just a regular VQA model with our component attached to it without relying on a specific loss function, so the usual losses that are used in VQA can be used directly. This allows our component to be easily used in other VQA models unlike their method.
- Their method has the advantage of interpretability of the outputs. To understand a predicted count one can look at the *set of objects* it counted (this is something that our Reviewer3 wanted). Our method has the advantage of interpretability of the model weights and activations. To understand a predicted count one can look at the *activations through the component* with a clear interpretation for the activations, i.e. understanding *how* the model made the decision (though unlike their method, without a set of objects being obtained in the process, but a score for each object between 0 and 1).
- In terms of performance, we can make some very rough comparisons with their numbers. The UpDown baseline that they re-implement is the same model architecture as used for the single-model results in our Table 1 by (Teney et al., 2017). This baseline model gets 47.4% accuracy on their dataset, which improves to 49.7% with their method (2.3% absolute improvement). Meanwhile, on number questions (a superset of counting questions, though mostly consisting of counting questions) with the regular VQA dataset, the same model architecture gets 43.9%, which improves to 51.4% with our model, a clearly much larger benefit (7.5%). Part of this is due to a stronger base model, but even then, the stronger baseline we compare against has a number accuracy of 46.6%,meaning that we have an absolute improvement of 4.8% with our model.
The improvement through our model on just the counting questions is even larger as we show in Table 2.

---

### Author Response · Authors · 2018-01-04
**Summary of rebuttal to main concerns**

Due to the length of our detailed point-by-point rebuttals, we would like to give a quick summary of our responses to the main concerns that the reviewers had.

# Reviewer 3 (convinced by our rebuttal and increased the rating)

- Too handcrafted
The current state-of-art in VQA on real images is nowhere near good enough for learning to count to be feasible using general models without hand-crafting some aspects of them for counting specifically. Hand-crafting gives the component many of its useful properties and guarantees, such as allowing us to understand why it made specific predictions. We are the first to use a graph-based approach with any significant benefit on VQA for real images, which in the future could certainly be generalized, but we had no success with generalizations so far.

- NMS baseline missing
We have updated the paper with NMS results, strengthening our main results.

- Comparison with (Chattopadhyay et al., 2017) missing
Their experimental setup majorly disadvantages VQA models, so their results are not comparable. We have updated the paper to make this clearer.


# Reviewer2 (No response to rebuttal yet)

- Application range is very limited
While the reviewer claims that our component is entirely limited to VQA, this is not true since even the toy dataset that we use has not much to do with VQA -- it is a general counting task. Counting tasks have much practical use and we updated the paper to explicitly state how the component is applicable outside of VQA.

- Built on a lot of heuristics without theoretical consideration
Our component does not use an established mathematical framework (such as Determinantal Point Processes as the reviewer suggests in a comment) but we are justifying every step with what properties are needed to count correctly. That is, we are mathematically correctly modelling a sensible counting mechanism and disagree with the claim that these are just a bunch of heuristics. In both theory and practice, the counting mechanism gives perfect answers when ideal case assumptions apply and sensible answers when they do not apply. The lack of traditional theory also seems to be a complaint about large parts of Deep Learning and recent Computer Vision research in general.

Especially given the strong positives that this reviewer lists, we do not think that a final rating of 4 is fair towards our work.


# Reviewer1 (No response to rebuttal yet)

- Fails to show improvement over a couple of important baselines
We think that the reviewer must have misunderstood something; we do not know what this could possibly be referring to. If the reviewer is referring to baselines in the paper, all results show a clear improvement of our component over all existing methods. If the reviewer is referring to baselines not in the paper, then we do not see how this can be the case: we only left out baselines that are strictly weaker in all aspects than the ones we show in the paper. You can verify that our results on the number category (51.39%) outperforms everything, including ensemble models of state-of-the-art techniques with orthogonal improvements, on the the official leaderboard: https://evalai.cloudcv.org/web/challenges/challenge-page/1/leaderboard (our results are hidden for anonymity)

- Qualitative examples of A, D, and C are needed
We have updated the paper to include some qualitative examples.

---

### Author Response · Authors · 2018-02-22
**Code release**

We have released our code at https://github.com/Cyanogenoid/vqa-counting

---

### Decision · Program_Chairs · 2018-01-29
**ICLR 2018 Conference Acceptance Decision**

**Decision:**

Accept (Poster)

**Comment:**

Initially this paper received mixed reviews. After reading the author response, R1 and and R3 recommend acceptance.

R2, who recommended rejecting the paper, did not participate in discussions, did not respond to author explanations, did not respond to AC emails, and did not submit a final recommendation. This AC does not agree with the concerns raised by R2 (e.g. I don't find this model to be unprincipled).

The concerns raised by R1 and R3 were important (especially e.g. comparisons to NMS) and the authors have done a good job adding the required experiments and providing explanations.

Please update the manuscript incorporating all feedback received here, including comparisons reported to the concurrent ICLR submission on counting.